# HetSSNet: Spatial-Spectral Heterogeneous Graph Learning Network for Panchromatic and Multispectral Images Fusion

**Mengting Ma** [* 1]   **Yizhen Jiang** [* 2]   **Mengjiao Zhao** [2]   **Jiaxin Li** [3]   **Wei Zhang** [2]

## Abstract

Remote sensing pansharpening aims to reconstruct spatial-spectral properties during the fusion of panchromatic (PAN) images and low-resolution multi-spectral (LR-MS) images, finally generating the high-resolution multi-spectral (HR-MS) images. In the mainstream modeling strategies, i.e., CNN and Transformer, the input images are treated as the equal-sized grid of pixels in the Euclidean space. They have limitations in facing remote sensing images with irregular ground objects. Graph is the more flexible structure, however, there are two major challenges when modeling spatial-spectral properties with graph: *1) constructing the customized graph structure for spatial-spectral relationship priors*; *2) learning the unified spatial-spectral representation through the graph*. To address these challenges, we propose the spatial-spectral heterogeneous graph learning network, named **HetSSNet**. Specifically, HetSSNet initially constructs the heterogeneous graph structure for pansharpening, which explicitly describes pansharpening-specific relationships. Subsequently, the basic relationship pattern generation module is designed to extract the multiple relationship patterns from the heterogeneous graph. Finally, relationship pattern aggregation module is exploited to collaboratively learn unified spatial-spectral representation across different relationships among nodes with adaptive importance learning from local and global perspectives. Extensive experiments demonstrate the significant superiority and generalization of HetSSNet.

---

[*]Equal contribution [1]School of Software Technology, Zhejiang University, Hangzhou, China [2]School of Computer Science and Technology, Zhejiang University, Hangzhou, China [3]School of Computer Science and Technology (National Exemplary Software School), Chongqing University of Posts and Telecommunications, Chongqing, China. Correspondence to: Wei Zhang <cstzhang-wei@zju.edu.cn>.

*Proceedings of the 42nd International Conference on Machine Learning*, Vancouver, Canada. PMLR 267, 2025. Copyright 2025 by the author(s).

## 1. Introduction

With the increasing demand of earth observation and monitoring, existing optical satellites (e.g., GaoFen-2, Quick-Bird) can simultaneously record bundled low-resolution multispectral (LR-MS) and high-resolution panchromatic (PAN) images from the same scene (Ma et al., 2025b;a). Due to physical limitations of existing satellite sensors, the recorded LR-MS image usually includes rich spectral property but relatively sparse spatial property, while their corresponding PAN image contains abundant spatial property but sparse spectral property. Since the remote sensing image with rich spatial-spectral properties, i.e., high-resolution multispectral (HR-MS) image, are crucial for practical applications (Li et al., 2022; Han et al., 2024; Asokan & Anitha, 2019; Cheng & Han, 2016; Li et al., 2022), pansharpening technique, which could obtain the HR-MS image by reconstructing spatial-spectral properties during the fusion of the recorded PAN and LR-MS images, has been widely explored (Xing et al., 2023).

Conventional pansharpening methods include component substitution (CS) approaches (Carper et al., 1990), multi-resolution analysis (MRA) methods (Khan et al., 2008), and variational optimization (VO) methods (Deng et al., 2019). These methods only have the ability of shallow non-linear representation; therefore, it is difficult to achieve a good performance of spatial and spectral properties reconstruction. Recently, pansharpening has witnessed significant progress with the development of deep learning. However, it is taken for granted that the mainstream learning-based models propose to process the input LR-MS and PAN images as a regular grid of pixels in the Euclidean space, i.e., treat all pixels of LR-MS and PAN images features in a fairly rigid manner. For example, in the CNN-based methods (e.g., DCFNet (Wu et al., 2022) and MDCUN (Yang et al., 2022)), each pixels of PAN and LR-MS images is rigidly designated to communicate with its nearest neighbors; in the Transformer-based model (e.g., HyperTransfromer (Bandara & Patel, 2022) and CTINN (Zhou et al., 2022b)), all pixels of LR-MS and PAN images are assigned to equal-sized attention grids for atten-

tion operations, trying to model all potential relationships between PAN and LR-MS images.

Remote sensing images consist of relatively stable ground objects, such as oceans, forests, buildings, and streets, etc. (Duan et al., 2024). Since these ground objects are usually not quadrate whose shape is irregular, the commonly-used grid structures in the modeling architecture like CNN and Transformer are redundant and inflexible to process them. In contrast, the graph structure is the more flexible topology (Han et al., 2022) to represent complex relationships (i.e., graph edges) between different attributes (i.e., graph nodes). However, there are two challenges to model spatial-spectral relationships with graph: (i) designing the customized graph structure in non-Euclidean space, which describes pansharpening-specific relationships priors, i.e., spatial relationship of PAN image, the intra-spectra relationship of LR-MS image, and the spectral relationship between LR-MS and PAN images, as analyzed in Sec. B of Appendix; (ii) modeling spatial and spectral relationships with one graph structure, which facilitates maintaining the balance of reconstructed spatial-spectral properties.

To address the aforementioned challenges, we propose the spatial-spectral heterogeneous graph learning network (**HetSSNet**) for remote sensing pansharpening that learns unified spatial-spectral relationships with one customized graph. Specifically, the spatial-spectral heterogeneous graph is first constructed for pansharpening, named **HetSS-Graph**, which consists of multiple heterogeneous nodes and edges, allowing the provided spatial-spectral relationship priors to be clearly described in the HetSS-Graph. Then, we design the basic relationship pattern generation module to extract the multiple spatial-spectral relationship patterns between nodes in HetSS-Graph. From the local and global perspectives, we design the spatial-spectral relationship aggregation module, which can aggregate the local information with adaptive importance learning of all basic relationship patterns, as well as learn global relevant information according to the similarity of global-wise relationship patterns between graph nodes. HetSSNet presents the first exploration in learning unified representations from multiple pansharpening-specific relationships for spatial-spectral properties reconstruction.

- We construct the first spatial-spectral heterogeneous graph structure, for remote sensing pansharpening. To the best of our knowledge, it's the first attempt for the potential of modeling pansharpening-specific spatial-spectral relationships in non-euclidean space.

- We customize a basic relationship generation module, for extracting multiple base spatial-spectral relationship patterns.

- Based on the extracted base relationship patterns,

we propose the relationship pattern aggregation module from the local and global perspectives, so that our model adaptively learn unified representation for spatial-spectral properties reconstruction.

- Extensive experimental results on three datasets show that our HetSSNet not only achieves favorable performance against previous SOTA approaches, but also generalizes well in real-world full-resolution scenes.

## 2. Related Work

### 2.1. Learning-based pansharpening method

In recent years, deep learning have dominated the remote sensing pansharpening community. These techniques leverage the powerful feature learning and nonlinear fitting capabilities inherent in neural networks, significantly outperforming traditional approaches. The focus of pansharpening is to model the spatial-spectral relationship during the fusion of PAN and LR-MS images. Recently, CNN-based models like PanNet (Yang et al., 2017), SRPPNN (Cai & Huang, 2020; Li et al., 2024; 2023a;b)and DCFNet (Wu et al., 2022) treat PAN and LR-MS images as a regular grid of pixels. In Transfromer-based models like HyperTransformer (Bandara & Patel, 2022) and CTINN (Zhou et al., 2022a), all pixels of feature maps extracted from PAN and LR-MS images are assigned to equal-sized attention grids for attention operations, trying to model all potential relationships between PAN and LR-MS images. Since the CNN and Transformer rigidly model relationships in Euclidean space, it is sub-optimal for remote sensing images with irregular objects. Graph is the more flexible modeling structure. In this paper, we construct the first spatial-spectral heterogeneous graph for pansharpening, which models the spatial-spectral relationships in non-Euclidean space.

### 2.2. Graph Representation Learning

GCN eases the assumption of prior conditions, which takes the research object as the node and the correlation or similarity between objects as the edge. It can deal with complex paired interactions and integrate global spatial data, make full use of the internal relations between objects, and mine invisible relations between objects. In recent years, the graph convolution theory has developed rapidly. It has not only been widely applied to various high-level vision tasks, such as action recognition (Zhao et al., 2019) and semantic segmentation (Li et al., 2020), (Qi et al., 2017), but also started to be used to solve low-level vision tasks, such as image inpainting (Wadhwa et al., 2021), image deraining (Fu et al., 2021), and image denoising (Valsesia et al., 2019). Furthermore, dual GCNs (Zhang et al., 2019) with different mapping strategies become popular. Bandara et al. (Bandara et al., 2022) proposed spatial and interaction space graph

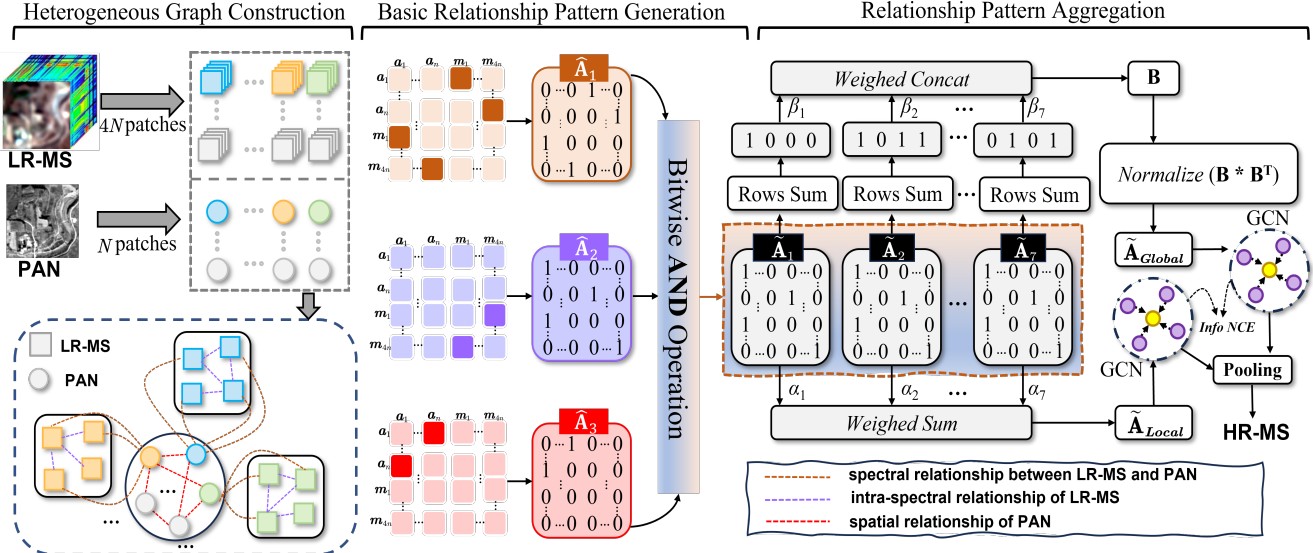

*Figure 1.* The overview of HetSSNet. Our HetSSNet consists of three components: spatial-spectral heterogeneous graph construction, basic relationship pattern generation and relationship pattern aggregation. According to the provided relationship priors, we construct the spatial-spectral heterogeneous graph structure in non-Euclidean space. Based on the constructed graph, we generate a series of basic spatial-spectral relationship pattern matrices, and finally aggregate these basic relationship patterns from the local and global perspectives.

reasoning to extract roads from aerial images. As far as we know, GCN is currently used for very little hyperspectral imagery. Qin et al. (Qin et al., 2018) and Wan et al. (Wan et al., 2019) have related work, but it is limited to the task of hyperspectral image classification (Hong et al., 2020; 2021; Kang et al., 2020). Recently, GPCNet (Yan et al., 2022) uses two independent GCN to directly model the spatial and spectral relationships of LR-MS and PAN image features' combination. In fact, it only uses GCN to model global relationships in Euclidean space. Our method designs graph structures in non-European space for pansharpening, and uses GCN to learn the graph node representations from both local and global perspectives.

## 3. Method

### 3.1. Problem Definition

*Definition 1 (Attributed Multiplex Heterogeneous Graph).* Given the defined graph $\mathcal{G} = (V, E, U)$, where V is a set of nodes, E is a set of observed edges. With the consideration of node and edge heterogeneity, we define the node type and edge type mapping functions as $\phi$: $V \rightarrow \mathbf{O}$ and $\psi$: $E \rightarrow \mathbf{R}$, where $\mathbf{O}$ and $\mathbf{R}$ denote the set of all node types and the set of all edge types, respectively. Each node $v \in V$ belongs to a particular node type, and each edge $e \in E$ is categorized into a specific edge type. if the U is the matrix that consists of node attribute features for all nodes, where each row is the associated node feature vector of node $v_i$, and $|\mathbf{O}| + |\mathbf{R}| > 2$, and there exist different types of edges

between same node pairs, the graph is called as ***Attributed multiplex heterogeneous graph***.

*Definition 2 (Basic Relationship Pattern).* The basic relationship pattern between two node types $\mathbf{O}_i$ and $\mathbf{O}_j$ in the heterogeneous graph is defined as $\mathbf{O}_i \xrightarrow{[r_1]\&[r_2]\&\cdots\&[r_{|\mathbf{R}|}]} \mathbf{O}_j$ which describes the complex relationships between two nodes, where $[\cdot]$ denotes optional, and at least one relation $r_i$ exists. Notice that for a heterogeneous graph with $\mathbf{R}$ relationships, it can generate up to $2^{|\mathbf{R}|} - 1$ basic relationship patterns. Taking the our constructed HetSS-Graph in Fig. 1 as an example, it has three types of edges, so it could generate up to 7 basic relationship patterns.

Based on the above definitions, we formally defined our studied problem for multiplex spatial-spectral relationships learning over the constructed heterogeneous graph.

### 3.2. Overall Architecture

As illustrated in Fig. 1, we present the details of our HetSS-Net with the overall architecture. Particularly, our HetSSNet consists of three key components: (i) spatial-spectral heterogeneous graph construction module (ii) Basic relationship pattern generation module, (iii) Relationship pattern aggregation module. Spatial-spectral heterogeneous graph (HetSS-Graph) structure is first constructed which explicitly describes the provided relationships priors for pansharpening. Basic relationship pattern generation module is used to extract all basic spatial-spectral relationship patterns from

the constructed heterogeneous graph, so as to make full use of the multiple spatial-spectral relationships in the heterogeneous graph. Relationship pattern aggregation module is designed to collaboratively learn node representations across different spatial-spectral relationships among nodes with adaptive importance learning from the local and global perspectives, finally fuse them to obtain the final output representation.

### 3.3. Spatial-Spectral Heterogeneous Graph Construction

The spatial-spectral heterogeneous graph structure (HetSS-Graph) is denoted as the attributed multiplex heterogeneous graph, which consists of multiple heterogeneous nodes and edges, explicitly describing pansharpening-specific relationship priors, i.e., spatial relationship of PAN image, intra-spectra relationship of LR-MS image and spectral relationship between PAN and LR-MS images, as shown in Fig. 1.

**Node in HetSS-Graph** *1) The first type.* Given a PAN image $P \in \mathbb{R}^{H \times W \times 1}$, we divide it into $N$ overlapping patches. By transforming each patch of PAN image into a feature vector $x_i \in \mathbb{R}^d$, we have $X_P = [x_1, x_2, \cdots, x_N]$ where $d$ is the feature dimension and $i = 1, 2, \cdots, N$. These features could be viewed as a set of unordered nodes which are denoted as $V_P = \{v_1, v_2, \cdots, v_N\}$. *2) The second type.* Given a LR-MS image $L \in \mathbb{R}^{H \times W \times 4}$, we also divide it into $N$ overlapping patches. For the LR-MS image patch $N_i \in \mathbb{R}^{\frac{H}{N} \times \frac{W}{N} \times 4}$, we transform each spectral band into a feature vector $y_i^b \in \mathbb{R}^d, b = 1, 2, 3, 4$, and we have $Y_i = \{y_i^1, y_i^2, \cdots, y_i^4\}$ where $d$ is the feature dimension and $i = 1, 2, \cdots, N$. These features can be viewed as the second type of nodes which are denoted as $V_L = \{v_i^1, v_i^2, \cdots, v_i^4\}$.

**Edge in HetSS-Graph** *1) The first type.* For the first type of node $v_i$, we use the $k$-nearest neighbor algorithm to find $k$ nodes that have the most similar features to that node, and connect edges between node $v_i$ and these neighboring nodes set $\mathcal{N}(v_i)$, and add the first type of edge directed from $v_j$ to $v_i$ for all $\mathcal{N}(v_i)$. The first type of edge establishes the spatial relationship of PAN image. *2) The second type.* For the second type of node $v_i^b$, we find its $k$ nearest neighbors $\mathcal{N}(v_i^b)$ and add the second type of edge directed from $v_j^b$ to $v_i^b$ for all $\mathcal{N}(v_i)$. The second type of edge establishes the intra-spectra relationship of LR-MS image. *3) The third type.* Given the first type of node $v_i$ and the second type of node $v_i^b$, we add the third type of edge directed from $v_i^b$ for all $\mathcal{N}(v_i^b)$ to $v_i$ for all $\mathcal{N}(v_i)$. The third type of edge establishes the spectral relationship between LR-MS image and PAN image. In particular, the each added edge is measured by cosine similarity.

### 3.4. Basic Relationship Pattern Generation Module

To make full use of the complex spatial-spectral relationships between nodes in the constructed heterogeneous graph, we first design a basic relationship pattern generation module that could directly extract all the basic spatial-spectral relationship pattern matrices from the constructed HetSS-Graph.

We first decouple the heterogeneous graph according to the type of edges. Let $\{A_r \in \mathbb{R}^{5n \times 5n} | r = 1, 2, 3\}$ denote the basic adjacency matrices, where $n$ is the number of all nodes in the graph. Then, each adjacency matrix and a corresponding logical variable (i.e., 0 or 1) are operated with the **XNOR** to generate 3 intermediate matrices $\{\hat{A}_r \in \mathbb{R}^{5n \times 5n} | r = 1, 2, 3\}$. Here, a logical variable of 1 is taken if the relation represented by the adjacency matrix is preserved in the basic spatial-spectral relationship pattern, and 0 otherwise. Finally, the intermediate matrices are bitwise **AND** operated to obtain the basic spatial-spectral relationship pattern matrices. That is, these relations corresponding to a logical variable value of 1 are retained in the final basic spatial-spectral relationship pattern. Notice that if a zero matrix is obtained, then this basic spatial-spectral relationship pattern does not exist in the graph. By adjusting the logical variables, all the basic spatial-spectral relationship pattern matrices $\{\tilde{A}_r \in \mathbb{R}^{5n \times 5n} | r = 1, 2, ..., 7\}$ can be obtained.

### 3.5. Relationship Pattern Aggregation Module

To automatically capture local information and global relevant information across different complex relationship pattern between nodes, we design the relationship pattern aggregation module from the local and global perspective.

**Local-wise Aggregation.** The overall process of local-wise aggregation is shown in the Fig. 1. After obtaining basic spatial-spectral relationship patterns, the local-wise aggregation aims to aggregate features from the complex structures among nodes from the local perspective by differentiating each basic spatial-spectral relationship pattern with different importance.

Specifically, the local-wise aggregation first uses a set of learnable weight parameters $\alpha_r$ to aggregate basic spatial-spectral relationship patterns as:

$$\tilde{A}_{Local} = \sum_{r=1}^{\mathbb{N}} \alpha_r \tilde{A}_r, \qquad (1)$$

where $\mathbb{N}$ is the number of obtained basic spatial-spectral relationship patterns. Then we feed the aggregated matrix $\tilde{A}_l$ into the graph convolution network (GCN). Following MHGCN (Yu et al., 2022b), our convolution also adopts the idea of simplifying GCN, that is, no nonlinear activation function is used. The single-layer GCN can effectively learn

the node representation that contains interaction information in all basic spatial-spectral relationship patterns. The $l$-layer node representation could been denoted as:

$$\mathbf{H}^l_{Local} = \tilde{\mathbf{A}}_{Local} \cdot \mathrm{U} \cdot \underbrace{\mathbb{W}^1_{Local} \cdots \mathbb{W}^l_{Local}}_{l}, \quad (2)$$

where U is the node attribute matrix, where each row is the associated node feature vector of $\mathrm{V_P}$ and $\mathrm{V_L}$. $\mathbb{W}^i_{Local}$ is the learnable weights of $i_{\mathrm{th}}$ layer, $i = 1, 2, \cdots, l$. We finally fuse outputs of all layers to obtain the local-wise node representation as:

$$\mathbf{H}_{Local} = \frac{1}{l} \sum_{i=1}^{l} \mathbf{H}^l_{Local}. \quad (3)$$

**Global-wise Aggregation.** The purpose of global-wise aggregation is to aggregate the features between nodes from the global perspective based on the similarity of global spatial-spectral relationship patterns between nodes. The overall process of global-wise aggregation is shown in the Fig. 1.

Specifically, we first generate the matrix to represent the global spatial-spectral relationship pattern of nodes based on the obtained basic spatial-spectral relationship patterns. In particular, we first add the rows to get a column vector for each basic spatial-spectral relationship pattern matrix. Here, each column vector describes the number of corresponding basic spatial-spectral relationship patterns of all nodes relative to all other nodes. Because different basic spatial-spectral relationship patterns may have different contributions to the similarity when calculating the similarity of global spatial-spectral relationship patterns, we use a set of learnable weights $\beta_i$ to concatenate pattern-specific column vectors to obtain the global spatial-spectral relationship pattern matrix $\mathbf{B} \in \mathbb{R}^{n \times \mathbb{N}}$ as:

$$\mathbf{B}_{m(i)} = \sum_{j=1}^{|\mathrm{V}|} \tilde{\mathbf{A}}_{m(i,j)}, \quad (4)$$

$$\mathbf{B} = Concat\left(\mathbf{B}_1 \cdots \mathbf{B}_{\mathbb{N}}\right) \cdot \triangle_\beta, \quad (5)$$

where $\mathbf{B}_m$ is the column vector corresponding to the $m$-th basic spatial-spectral relationship pattern, $\triangle_\beta = \mathbf{Diag}(\beta_1, \beta_2, \cdots, \beta_{\mathbb{N}})$ is the learnable diagonal matrix. Then we multiply the global spatial-spectral relationship pattern matrix by its transpose and normalize it to obtain the global spatial-spectral relationship pattern similarity matrix:

$$\tilde{\mathbf{A}}_{Global} = \mathrm{normalize}(\mathbf{B} \cdot \mathbf{B}^T). \quad (6)$$

Intuitively, the more similar the global spatial-spectral relationship pattern of two nodes is, the greater their weight in the similarity matrix is. Next, we input the global spatial-spectral relationship pattern similarity matrix into the GCN for information aggregation, so as to obtain the node representation from the global spatial-spectral relationship pattern similarities:

$$\mathbf{H}^l_{Global} = \tilde{\mathbf{A}}_{Global} \cdot \mathrm{U} \cdot \underbrace{\mathbb{W}^1_{Global} \cdots \mathbb{W}^l_{Global}}_{l} \cdot \quad (7)$$

The last layer output $\mathbf{H}^l_{Global}$ is denoted as the global node representation $\mathbf{H}_{Glocal}$.

**The Final Output.** Contrastive learning has demonstrated its superiority in various graph learning tasks. Inspired by that, we propose to use contrastive learning to empower the representation learning ability of the model that maximizes the agreement of node representations learned from a local perspective and a global perspective.

$$\mathcal{L}_{cl} = -\sum_{i \in \mathrm{V}} \log \frac{\exp\left(s(\mathbf{H}_{Local,i}, \mathbf{H}_{Global,i})/\tau\right)}{\sum_{j \in \mathrm{V}} \exp(s(\mathbf{H}_{Local,i}, \mathbf{H}_{Global,j})/\tau)}, \quad (8)$$

where $\mathbf{H}_{*,i}$ is the local-wise/global-wise node representation of the $i$-th node, $s(\cdot, \cdot)$ denotes the cosine similarity function, and $\tau$ is the tunable temperature hyperparameter to adjust the scale for softmax. This contrastive learning allows the local spatial-spectral relationship pattern and global spatial-spectral relationship pattern to collaboratively supervise each other, which enhances the node representation learning. Eventually, we use the outputs of both the local-wise aggregation and the global-wise aggregation to obtain the final node representation $\mathbf{H} \in \mathbb{R}^{n \times d}$ through the average pooling operation for downstream tasks as:

$$\mathbf{H} = \frac{1}{2}(\mathbf{H}_{Local} + \mathbf{H}_{Global}). \quad (9)$$

### 3.6. Optimization

We adopt the $L_1$ loss function, which ensures that the network output $\mathcal{H}$ is as close as possible to the corresponding ground truth image GT.

$$\mathcal{L}_1 = \|\mathcal{H} - \mathrm{GT}\|_1 \quad (10)$$

Finally, we integrate $L_1$ loss with our contrastive learning loss to optimize our model jointly:

$$\mathcal{L} = \mathcal{L}_1 + \gamma \mathcal{L}_{cl} \quad (11)$$

where $\gamma$ is the hyperparameter for tuning the importance of contrastive learning.

## 4. Experiments

### 4.1. Dataset and Benchmark

*1) Dataset:* We conduct experiments using the widely recognized WorldView-3, QuickBird and GaoFen-2 datasets (Ma et al., 2024). The WorldView-3 dataset consists of instances acquired by the sensor aboard the WorldView-3 satellite. This sensor captures data which covers wavelengths from 0.4 to 1 $\mu$m, with a spatial resolution of 1.24 m. The Quick-Bird dataset consists of instances acquired by the sensor

| Datasets | GaoFen-2 | QuickBird | WorldView-3 |
|---|---|---|---|
| Raw image | 7 | 5 | 3 |
| Bit depth | 11 | 10 | 10 |
| Training set | 35725 | 12119 | 11856 |
| Testing set | 3370 | 356 | 3639 |
| LR-MS image size | $32 \times 32 \times 4$ | $32 \times 32 \times 4$ | $32 \times 32 \times 4$ |
| PAN image size | $128 \times 128 \times 1$ | $128 \times 128 \times 1$ | $128 \times 128 \times 1$ |
| Target HR-MS image size | $128 \times 128 \times 4$ | $128 \times 128 \times 4$ | $128 \times 128 \times 4$ |

*Table 1.* The detailed dataset information (GaoFen-2, WorldView-3 and QuickBird).

aboard the QuickBird satellite. This sensor captures data across four spectral bands, covering wavelengths from $0.45$ to $0.9$ $\mu$m, with a spatial resolution of $2.4$ m. The images in the GaoFen-2 dataset are collected by the sensor onboard the GaoFen-2 satellite, which records data across four spectral bands within the wavelength range of $0.45 - 0.89$ $\mu$m. Additionally, this sensor provides a spatial resolution of $3.2m$. The data generation process adheres strictly to Wald's protocol (Wald et al., 1997), with comprehensive details provided in (Deng et al., 2022). As shown in Tab. 1, we present the detailed information of traning dataset and testing dataset in the experiment.

*2) Benchmark:* We compare our method with the following two groups of methods: **(1) Learning-based methods:** SRPPNN (Cai & Huang, 2020), DCFNet (Wu et al., 2022), CTINN (Zhou et al., 2022b), Hyperfomer (Bandara & Patel, 2022), SFIINet (Zhou et al., 2022c), BiMPan (Hou et al., 2023), MDCUN (Yang et al., 2022), MSDDN (He et al., 2023), LGTEUN (Li et al., 2023c), FAMENet (Xuanhua et al., 2024), GPCNet (Yan et al., 2022), FusionMamba (Peng et al., 2024), WINet (Tan et al., 2024) and HFIN (Zhang et al., 2024). **(2) Traditional methods:** SFIM (Liu, 2002), GS (Sandhu et al., 2021) and BROVEY (Gillespie et al., 2003). ***All comparison methods are re-trained on the adopted datasets, without directly using the experimental details in the original articles. A description of all comparison methods could be shown in the appendix.***

### 4.2. Evaluation Metrics

Following previous studies on pansharpening, five image quality assessment metrics (Yang et al., 2023) are employed for evaluation on reduced-resolution images, including spectral angle mapper (SAM), dimensionless global error in synthesis (ERGAS), the structural similarity (SSIM), the peak signal-to-noise ratio (PSNR), and the spatial correlation coefficient (SCC) (Zhou et al., 1998). Specifically, PSNR measures the ratio between peak signal and noise in the image, reflecting the distortion level of the fusion result. SSIM and SCC effectively measure the spatial similarity of the results. SAM and ERGAS are used to measure angular and dynamic range differences between the fused image and the ground truth image, respectively. Additionally, to further

assess the generalization ability of our method, we test it on the corresponding full-resolution real-world scene. Since there are not GT images available for the full-resolution dataset, we use three non-reference metrics to evaluate the performance of the model: Spectral Distortion Index ($D_\lambda$), Spatial Distortion Index ($D_s$), and No-Reference Quality (QNR) (Alparone et al., 2008). $D_\lambda$ concerns spectral shifts between LR-MS and fused HR-MS images, while $D_s$ considers spatial disparities between PAN and HR-MS images. QNR is a composite metric measuring spatial and spectral deviations. In these metrics, ideal values for SAM, ERGAS, $D_s$, and $D_\lambda$ are 0, while higher values indicate better model performance for the remaining metrics.

### 4.3. Training Details

During the training of our networks on the Wordview-3, QuickBird, and GaoFen-2 datasets, the number of iterations is set to 30000, 28000, and 28000, respectively . The Adam optimizer with $\beta_1 = 0.9$ and $\beta_2 = 0.999$ is employed for the optimization, and the batch size is set as 4. Our proposed method is trained for 30000 iterations. The initial learning rate is set to $1 \times 10^{-4}$, and decays by 0.85 after every 3000 iterations. Additionally, all the experiments are conducted in the PyTorch framework with four NVIDIA RTX A6000 GPUs.

### 4.4. Experiment Results

In Tab. 2, we report the average evaluation metrics of our proposed method and other selected algorithms on the three datasets, i.e., WorldView-3, QuickBird, and GaoFen-2. The best and second-best results are highlighted in boldface and underlined, respectively. It is evident that our method outperforms previous methods in almost all metrics across all three datasets. In particular, all the evaluation metrics of the learning-based methods significantly outperform the traditional methods on the three adopted datasets, which is consistent with the subjective analysis mentioned above and aligns with our understanding. This is because traditional methods are often based on hand-crafted designs, limiting the capability to reconstruct spatial-spectral properties for target HR-MS images. The experimental results also demonstrate that HetSS-Graph excels in spatial texture reconstruction and spectral fidelity, as evidenced by its minimum SAM value and maximum SSIM and PSNR values. Specifically, CNN and Transfromer methods (e.g., DCFNet and Hyper-Transformer) generally perform poorly on the QuickBird dataset, which is because the rigid modeling architecture limits their ability to handle complex and irregular objects in remote sensing images; GPCNet directly use GCNs to model spatial/spectral relationship between PAN and LR-MS images features, breaking the balance between spectral property recovery and spatial property reconstruction. Our HetSSNet designs the graph structure in the Non-euclidean

| | WorldView-3 | | | | | QuickBird | | | | | GaoFen-2 | | | | |
|---|---|---|---|---|---|---|---|---|---|---|---|---|---|---|---|
| | PSNR↑ | SSIM↑ | SAM↓ | ERGAS↓ | SCC↑ | PSNR↑ | SSIM↑ | SAM↓ | ERGAS↓ | SCC↑ | PSNR↑ | SSIM↑ | SAM↓ | ERGAS↓ | SCC↑ |
| SFIM | 30.982 | 0.749 | 0.076 | 6.497 | 0.821 | 33.463 | 0.818 | 0.036 | 3.071 | 0.792 | 37.840 | 0.917 | 0.042 | 3.836 | 0.806 |
| BROVEY | 30.857 | 0.761 | 0.086 | 5.506 | 0.819 | 30.109 | 0.837 | 0.077 | 2.473 | 0.739 | 36.046 | 0.902 | 0.036 | 2.261 | 0.835 |
| GS | 35.143 | 0.807 | 0.078 | 5.409 | 0.825 | 32.454 | 0.767 | 0.033 | 2.198 | 0.751 | 38.328 | 0.930 | 0.037 | 2.826 | 0.852 |
| SRPPNN | 38.642 | 0.887 | 0.066 | 5.244 | 0.904 | 33.966 | 0.886 | 0.082 | 2.840 | 0.875 | 40.918 | 0.961 | 0.036 | 2.683 | 0.911 |
| DCFNet | 34.429 | 0.862 | 0.098 | 6.373 | 0.568 | 34.661 | 0.912 | 0.072 | 2.573 | 0.848 | 47.055 | 0.988 | 0.027 | 1.270 | 0.912 |
| CTINN | 39.497 | 0.889 | 0.063 | 5.064 | 0.962 | 35.705 | 0.898 | 0.038 | 2.422 | 0.922 | 46.875 | 0.987 | 0.023 | 1.302 | 0.960 |
| SFIINet | 39.026 | 0.885 | 0.064 | 5.248 | 0.643 | 34.459 | 0.893 | 0.041 | 2.811 | 0.912 | 47.195 | 0.988 | 0.022 | 1.267 | 0.980 |
| Hyperformer | 34.121 | 0.843 | 0.098 | 6.356 | 0.867 | 34.374 | 0.869 | 0.072 | 2.555 | 0.914 | 45.499 | 0.981 | 0.034 | 1.584 | 0.959 |
| MDCUN | 39.929 | 0.892 | 0.056 | 4.864 | 0.970 | 36.070 | 0.891 | 0.036 | 2.377 | 0.910 | 48.668 | 0.991 | 0.018 | 1.068 | _0.986_ |
| BiMPan | 38.955 | 0.879 | 0.069 | 5.253 | 0.925 | 37.162 | 0.919 | 0.032 | 2.045 | 0.931 | 45.901 | 0.984 | 0.026 | 1.465 | 0.974 |
| LGTEUN | _40.124_ | _0.895_ | _0.055_ | 4.791 | **0.974** | 35.911 | 0.924 | 0.045 | 2.170 | 0.950 | 48.283 | _0.992_ | 0.022 | 1.126 | 0.979 |
| MSDDN | 35.127 | 0.840 | 0.085 | 6.232 | 0.766 | 34.751 | 0.872 | 0.038 | 2.538 | 0.935 | 39.902 | 0.942 | 0.039 | 2.883 | 0.886 |
| FAMENet | 37.219 | 0.883 | 0.071 | 5.737 | 0.902 | _37.218_ | _0.929_ | _0.030_ | **1.996** | **0.953** | 42.983 | 0.981 | 0.017 | 1.013 | 0.978 |
| GPCNet | 38.821 | 0.883 | 0.062 | 5.242 | 0.909 | 33.972 | 0.889 | 0.076 | 2.835 | 0.889 | 40.926 | 0.967 | 0.032 | 2.671 | 0.923 |
| FusionMamba | 38.727 | 0.872 | 0.069 | 5.437 | 0.934 | 34.021 | 0.881 | 0.047 | 2.931 | 0.904 | 46.242 | 0.976 | 0.027 | 1.278 | 0.972 |
| WINet | 39.781 | 0.887 | 0.056 | _4.769_ | 0.969 | 37.209 | 0.927 | 0.032 | 2.212 | 0.931 | _49.405_ | 0.984 | _0.016_ | _0.968_ | 0.977 |
| HFIN | 32.126 | 0.837 | 0.083 | 6.627 | 0.837 | 33.616 | 0.811 | 0.050 | 3.174 | 0.869 | 41.938 | 0.966 | 0.039 | 2.534 | 0.894 |
| **HetSSNet** | **40.623** | **0.905** | **0.054** | **4.741** | _0.972_ | **37.228** | **0.931** | **0.029** | _2.022_ | _0.951_ | **49.541** | **0.994** | **0.015** | **0.955** | **0.990** |

*Table 2.* The average quantitative results of reduced-resolution scenes over three datasets. Best results are highlighted in boldface, the second best one is underlined.

space, and learns a unified representation of spatial-spectral properties in the graph structure, achieving competitive results in all indicators.

The subjective evaluation results are shown in Fig. 2 for the WorldView-3 dataset, using Red-Green-Blue bands composition for presentation. The first two rows present the pan-sharpened images, while the third and fourth rows display the mean square error (MSE) maps between the pan-sharpened images and the HR-MS images. As evident from the results, our HetSSNet is capable of restoring more realistic textures than existing pansharpening methods and markedly retain the spectral property of ground objects. The traditional methods, i.e., SFIM, BROVEY, and GS struggle to represent details satisfactorily, leading to blurred building edges and spectral distortion. Similarly, the learning-based methods face challenges in generating sharp and clear textures as well. For instance, SRPPNN and MSDDN produce results with notable color inconsistency. Although MDCUN and FAMENet produce visually more promising results, their error values are still higher than ours. The HetSS-Net generates results that are closer to the target HR-MS images with higher perceptual quality and lower reconstruction errors. The visualization results on the QuickBird and GaoFen-2 scenes are shown in Sec. D of Appendix.

To demonstrate the generalization capability of the proposed method, we conduct quantitative evaluations of our pre-trained model on 20 unseen full-resolution (real-world) images, the corresponding results are presented in Sec. D of Appendix.

| Model | PSNR ↑ | SAM ↓ | SSIM ↑ | QNR ↑ | $D_s$ ↓ |
|---|---|---|---|---|---|
| (a) | 48.212 | 0.035 | 0.976 | 0.841 | 0.137 |
| (b) | 48.923 | 0.023 | 0.981 | 0.852 | 0.121 |
| (c) | 48.536 | 0.027 | 0.972 | 0.843 | 0.139 |
| (d) | 49.221 | 0.019 | 0.991 | 0.857 | 0.112 |
| Ours | 49.541 | 0.015 | 0.994 | 0.860 | 0.107 |

*Table 3.* Ablation study of base relationship pattern extraction and relationship pattern aggregation.

## 4.5. Ablation Studies

To evaluate the effectiveness of each component of our model, we further conduct the ablation studies on the GaoFen-2 dataset.

**Basic relationship pattern generation module.** As shown in Tab. 3, in Model (a), we replace the basic relationship pattern generation module with the meta-path sampling strategy (Fu et al., 2020). In Model (b), we replace the basic relationship pattern generation module with the decoupled adjacency matrices (Yu et al., 2022a). Compared Model (a) and Model (b), the performance of our HetSSNet drops significantly on the GaoFen-2 dataset. This also demonstrates the contribution of the proposed relationship pattern generation module to the model performance improvement.

**Relationship pattern aggregation module.** As shown in Tab. 3, in Model (c), we remove the local-wise relationship pattern aggregation module, that is, only global-wise relationship pattern aggregation module is kept. In Model

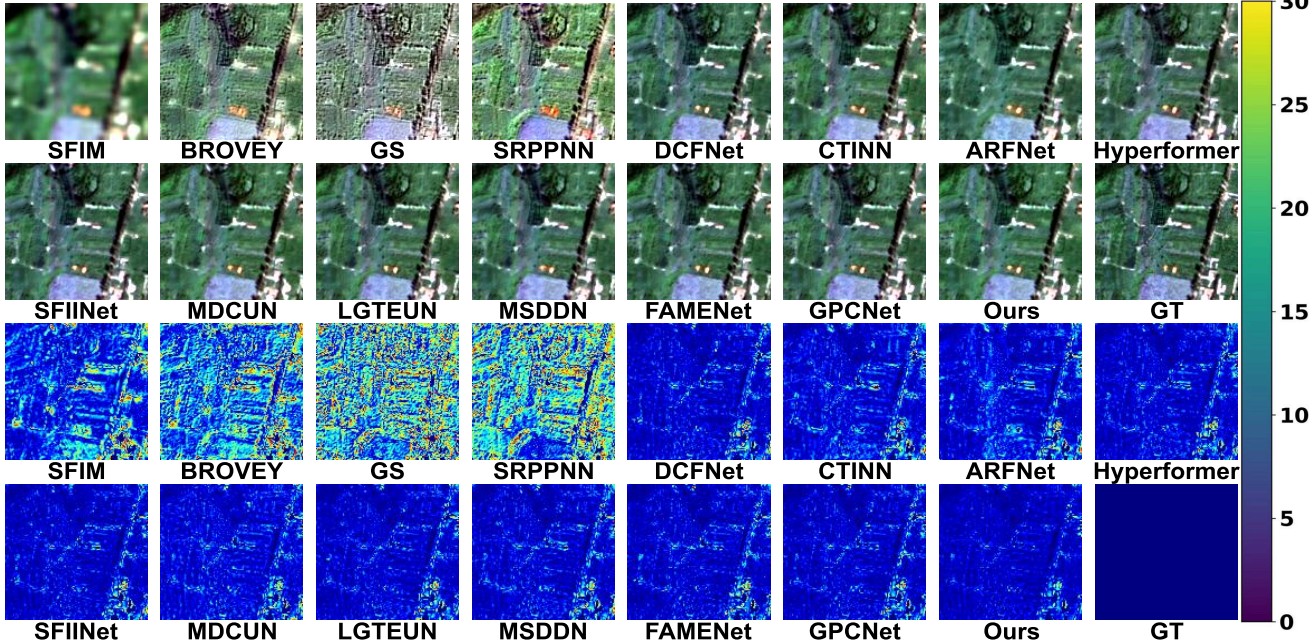

*Figure 2.* Qualitative results of reduced-resolution scene on the WorldView-3 dataset. Top group: the fused results. Bottom group: the error between fused results and reference.

(d), we only keep the local-wise relationship pattern aggregation module. Model (c) performs the worst and is significantly worse than Model (d). Local-wise aggregation is to aggregate local feature information, which is the intrinsic attribute of node representation and thus plays a decisive role in graph learning. Global-wise aggregation can pass feature information between nodes based on the similarity of spatial-spectral relationship patterns, effectively supplementing global relevant information to facilitate node representation. Model (c) and Model (d) are worse than the our method, which verifies that both play an important role in node representation learning of the HetSSNet for spatial-spectral properties reconstruction.

**The number of aggregation layer.** As shown in Tab. 4, firstly, the performance of HetSSNet increases with the increasing number of layers. When the number of layers reaches 2 to 3, the model has a significant performance drop on several metrics. It should be emphasized that the purpose of this work is not to solve the over-smoothing problem of GCN, but to aggregate the relevant global information more effectively. Thus, even when the number of layers is set too small, e.g., 2, our model can also learn global spatial-spectral relationship to improve model performance.

**Effect of hyperparameter** $\gamma$**.** From Tab. 5, we can find that our HetSS-Graph achieves the best performance on almost all datasets when $\gamma$ falls near 0.03. The model performance is improved on all datasets when $\gamma$ increases from 0 to 0.03, indicating that contrastive learning can effectively coordi-

| Layer | PSNR ↑ | SAM ↓ | SSIM ↑ | QNR ↑ | $D_s$ ↓ |
|-------|--------|-------|--------|-------|---------|
| $l=1$ | 47.683 | 0.041 | 0.967  | 0.836 | 0.145   |
| $l=2$ | 49.541 | 0.015 | 0.994  | 0.860 | 0.107   |
| $l=3$ | 49.227 | 0.018 | 0.991  | 0.856 | 0.112   |
| $l=4$ | 49.104 | 0.024 | 0.986  | 0.851 | 0.119   |

*Table 4.* Ablation study of aggregation layer.

| $\gamma$ | PSNR ↑ | SAM ↓ | SSIM ↑ | QNR ↑ | $D_s$ ↓ |
|----------|--------|-------|--------|-------|---------|
| 0        | 48.559 | 0.036 | 0.981  | 0.842 | 0.129   |
| 0.01     | 49.541 | 0.015 | 0.994  | 0.860 | 0.107   |
| 0.5      | 49.103 | 0.018 | 0.988  | 0.852 | 0.112   |
| 1        | 48.629 | 0.028 | 0.987  | 0.845 | 0.125   |

*Table 5.* Ablation study of the hyperparameter $\gamma$ shown in Eq. 11.

nate the node representation learning of the two aggregation modules in HetSSNet for improving graph node representation performance. When $\gamma$ is set too large (e.g., $\gamma = 1$), it will affect the importance of the $\mathcal{L}_1$ loss in the model learning, and thus damaging the reconstruction performance.

## 5. Conclusion

This paper presents HetSSNet, a novel spatial-spectral heterogeneous graph learning network dedicated to learning unified spatial-spectral representation for pansharpen-

ing. We first construct the first spatial-spectral heterogeneous graph structure to explicitly describe different pansharpening-specific relationship priors. Based on the constructed heterogeneous graph structure, we propose the base relationship pattern generation module to extract multiple spatial-spectral relationship patterns. Furthermore, we propose an relationship pattern aggregation module to learn local-wise and global-wise information among nodes Experimental results show that our HetSSNet outperforms existing SOTA methods and especially excels in reconstructing spatial-spectral properties. In the future, we will focus on evaluating our method under more challenging satellite image scenarios, e.g., there are fast-moving objects in the scenarios.

## Acknowledgements

This work was in part supported by the Science and Technology Innovation 2025 Major Project of Ningbo City (Grant No. 2023Z130).

## Impact Statement

This paper presents work whose goal is to advance the field of Machine Learning. There are many potential societal consequences of our work, none which we feel must be specifically highlighted here.

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

## A. Appendix Outline

In this supplementary material, we provide more details of our HetSSNet as follows:

- Sec. B conducts detailed analysis and reveals the complex relationship priors for pansharpening, as illustrated in Fig. 3 and Fig. 4,

- Sec. C describes more details about each benchmark.

- Sec. D reports more quantitative and visual results, i.e., visual results on reduced-resolution scene, quantitative results on full-resolution dataset, quantitative results on eight-band WorldView-3 dataset and visual results on full-resolution scene.

- Sec. E reports complexity analysis of our HetSSNet.

## B. Pansharpening-specific relationship priors

In this section, we reveal a spatial property-related relationship prior (**Prior 1**) and a spectral property-related relationship prior (**Prior 2**), as illustrated in Fig. 3 and Fig. 4. This analysis is performed on the widely-used GaoFen-2 dataset, which have 38645 sets of PAN, LR-MS and corresponding HR-MS (GT) images.

**Prior 1:** In order to reconstruct spatial property for target HR-MS image, spatial relationship of PAN need to be modeled.

**Analysis 1:** Fig. 3 compares the spatial distribution histograms between PAN image and GT image in terms of each of their spectral band, as well as LR-MS image and their GT images in terms of spectral band. As shown in each sub-graph of Fig. 3, the correlation coefficient measures the correlation between the two histograms, which is obtained by the EMD distance. From the first and second rows, we could obtain that the spatial distribution of PAN is more similar to the spatial distribution of GT image's each spectral band. Here, the spectral band is defined as $band_i$, where $i = 1, 2, 3, 4$. For example, the correlation coefficient between the PAN image and GT image's $band_1$ is $0.423$, while the correlation coefficient between LR-MS's $band_1$ and GT's $band_1$ is $0.023$. Similar results can be found for the remaining sub-graph of Fig. 3. Compared by LR-MS image, the spatial distribution of PAN image is much more similar to the spatial distribution of its target HR-MS (GT) image. Thus, we obtain the conclusion that only modeling spatial relationship of PAN image could reconstruct required spatial property for the target HR-MS image.

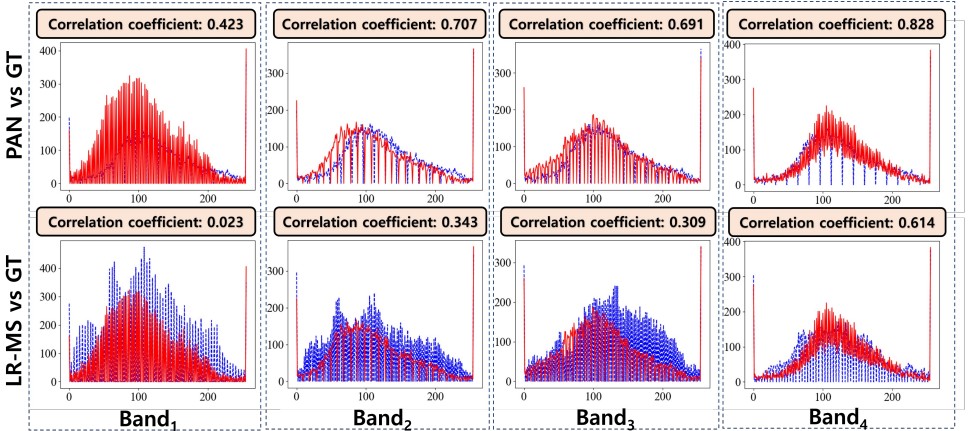

The distribution comparison between PAN (blue lines) / LR-MS (blue lines) and GT (red lines).

*Figure 3.* The spatial distribution comparison between LR-MS/PAN and GT (target HR-MS) in terms of each spectral band. The band illustrated in the figure refers to the spectral band. The correlation coefficient represents the correlation between the two distribution curves, and the larger the value, the greater the correlation.

**Prior 2:** In order to reconstruct spectral property for target HR-MS image, intra-spectra relationship of LR-MS, as well as the spectral relationship between PAN and LR-MS images, needs to be modeled.

**Analysis 2:** Here, we investigate the correlation between the LR-MS' spectral bands and the correlation between the GT's spectral bands. Fig. 4 visualizes their correlation and compares the distribution histograms between spectral bands (band$_1$ vs. band$_2$, band$_2$ vs. band$_3$ and band$_3$ vs. band$_4$) of the paired LR-MS and GT images. It can be observed that the distribution correlation between the LR-MS's spectral bands and distribution correlation between the GT's spectral bands is largely different. For example, the correlation coefficient between the GT's band$_3$ and GT's band$_4$ is $0.641$, while the correlation coefficient between the LR-MS's spectral bands (band$_3$ vs. band$_4$) is $0.263$. Similar results can be found for the remaining sub-graph of Fig. 4. The above analysis shows that it is sub-optimal that model only the LR-MS's spectral relationship. Thus, to reconstruct spectral property for target HR-MS image, the spectral relationship of LR-MS, as well as the spectral relationship between PAN and LR-MS images, needs to be modeled.

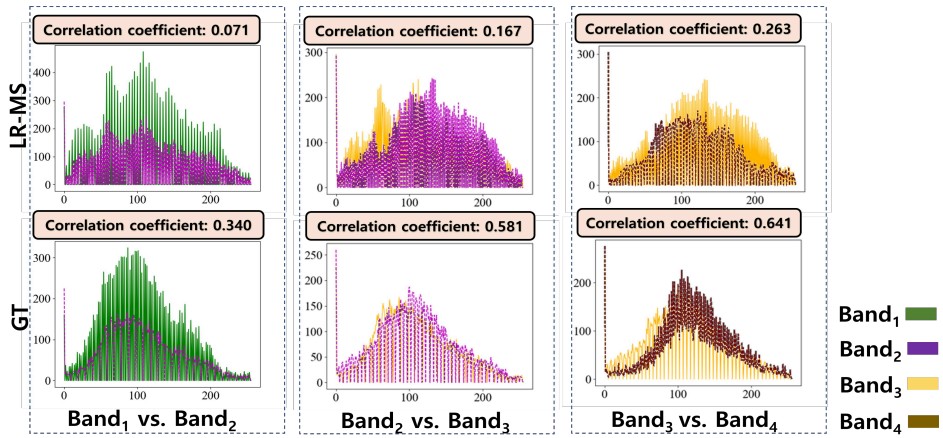

The distribution comparison between spectral bands of LR-MS/GT.

*Figure 4.* The distribution comparison between LR-MS's spectral bands, and the pixel distribution comparison between GT's spectral bands. The band illustrated in the figure denotes the spectral band. The correlation coefficient represents the correlation between the two distribution curves, and the larger the value, the greater the correlation.

## C. The benchmarks

**SRPPNN** proposes a pansharpening algorithm based on a deep convolutional neural network. It extracts intrinsic spatial details from multispectral images through a super-resolution process and combines progressive pansharpening and a high-pass residual module to enhance the spatial resolution of the images.

**DCFNet** is a dynamic cross-feature fusion network based on convolutional neural networks. It employs three parallel branches (high, medium, and low resolutions) for feature fusion to generate high-resolution multispectral images with enhanced spatial resolution and complete spectral information.

**CTINN** builds the Transformer-CNN dual-branch network, which uses CNN and Transformer to simultaneously extract the local and global features of PAN and LR-MS images' combination.

**Hyperformer** is based on transformer architecture and adopts the stacked multi-head attention to model the spatial dependencies between LR-MS and PAN image features.

**SFIINet** uses cascaded spatial and channel attention mechanisms to fuse local-global features encoded by two CNN-based branches.

**BiMPan** is based on CNN, and it introduces a bidomain modeling approach for pansharpening, combining local feature extraction in the spatial domain with global detail reconstruction in the Fourier domain to enhance the spatial resolution of multispectral images.

**MDCUN** is an interpretable deep neural network that formulates the pan-sharpening problem as a variational model minimization task. It employs a CNN-based iterative algorithm to construct the model, searching for similarities between long-range patches. Additionally, it combines PAN images with each band of MS images to selectively provide high-

frequency details.

**MSDDN** is a method based on CNN that enhances the effect of pansharpening by integrating multi-scale information from both the spatial domain and the frequency domain. Specifically, it employs a spatial guidance sub-network to learn local spatial information and a frequency guidance sub-network to learn global frequency-domain information.

**LGTEUN** is an interpretable deep unfolding network that alternately completes data updates in each iterative stage through a CNN-based data module and a Transformer-based prior module.

**FAMENet** is implemented based on CNN, where the Adaptive Frequency Separation Prediction Module utilizes the discrete cosine transform to achieve frequency separation by predicting a frequency mask. The Sub-Frequency Learning Expert Module is responsible for reconstructing high-frequency and low-frequency information. The Expert Mixture Module dynamically weights the high-frequency and low-frequency information described above.

**GPCNet** is implemented based on graph convolutional networks and enhances the model's representation capability by introducing asynchronous knowledge distillation. Specifically, it employs the spatial GCN module to capture global spatial information in the image, the spectral band GCN module to capture the global correlation of spectral features and enhance spectral fidelity, and the atrous spatial pyramid module to learn multi-scale feature information.

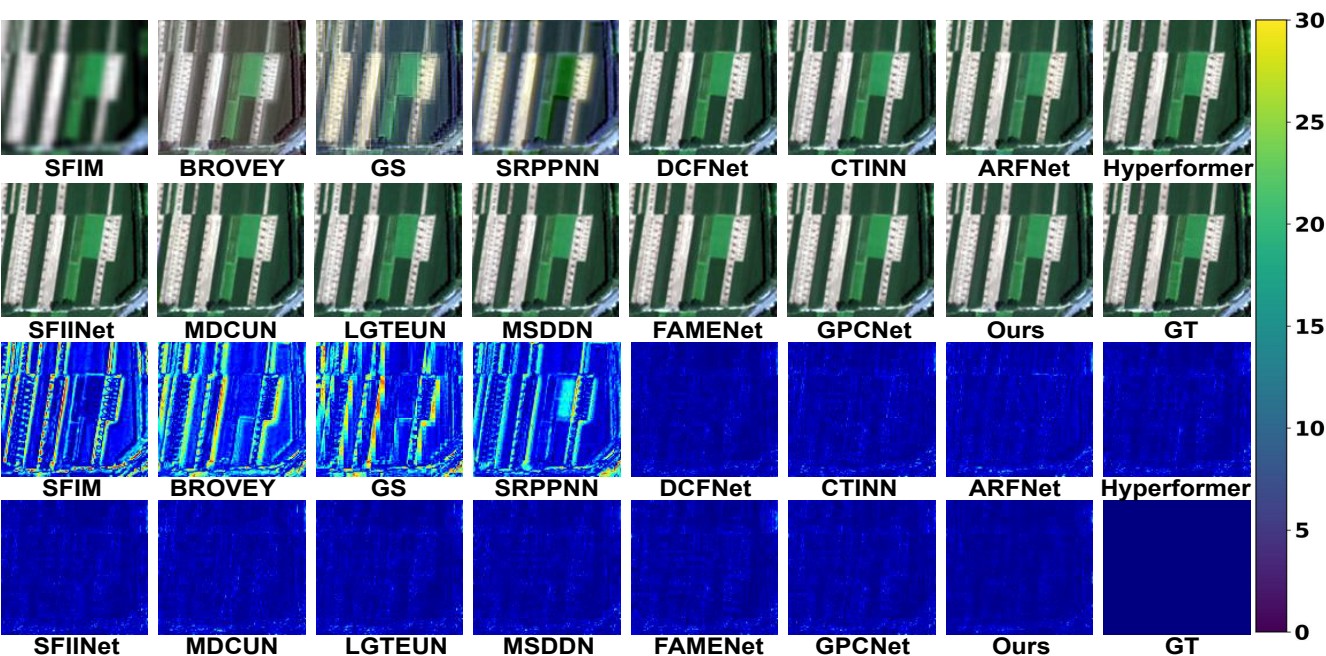

*Figure 5.* Qualitative results of reduced-resolution scene on the QuickBird dataset. Top group: the fused results. Bottom group: the error between fused results and reference.

## D. More quantitative and visualization results

As shown in Fig. 5 and Fig. 6, we present the visualization results of the QuickBird and GaoFen-2 datasets.

We present the results on the eight-band worldView-3 dataset, and the results prove that our method has obtained promising results, as shown in Tab. 7. To demonstrate the generalization capability of the proposed method, we conduct quantitative evaluations of our pre-trained model on 20 unseen full-resolution (real-world) images for each of the GaoFen-2, WorldView-3, and QuickBird datasets. Tab. 6 presents the results, and the best and second-best results are highlighted in boldface and underlined, respectively. We could see that the deep learning-based techniques still perform favorably against their traditional counterparts. In addition, our proposed framework achieves the optimal outcomes for all indexes, confirming its superior generalization capability compared to both traditional and deep learning-based pansharpening methods.

Additionally, we conduct qualitative analysis on the real-world scene of GaoFen-2 dataset. The visual comparison of several methods on a representative full-resolution test scene has been illustrated in Fig. 7. From the figure, it is evident

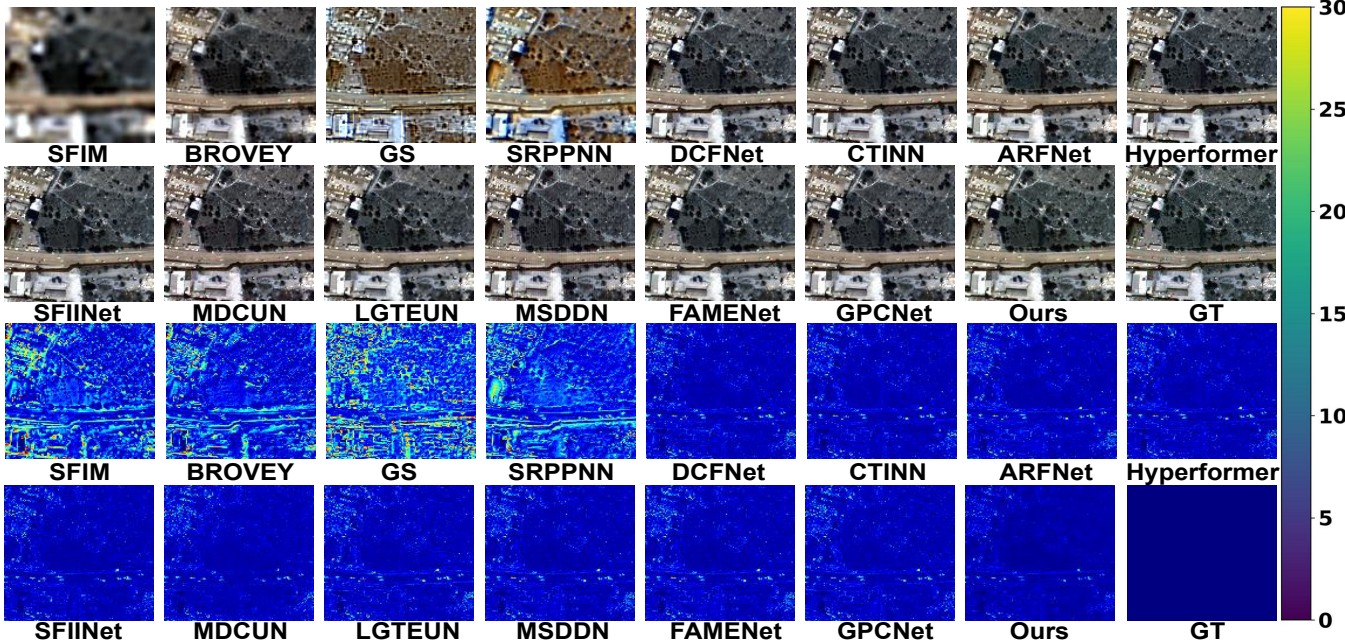

*Figure 6.* Qualitative results of reduced-resolution scene on the GaoFen-2 dataset. Top group: the fused results. Bottom group: the error between fused results and reference.

that traditional methods either suffer severe blurring, such as SFIM and GS, or exhibit pronounced spectral distortions, as seen with BROVEY. Even though deep learning has brought some spatial detail improvements to pansharpening models, modeling spectral properties has been a challenge for these models, exemplified by ARFNet, HyperTransformer, and MSDDN. Compared with the these CNN and Transformer-based methods, our method models the spatial-spectral relationship based on a non-Euclidean graph structure, which has advantages in reconstructing the spatial-spectral properties of complex objects. Overall, our method demonstrates outstanding performance in both quantitative and qualitative experiments on the full-resolution dataset, indicating its strong generalizability.

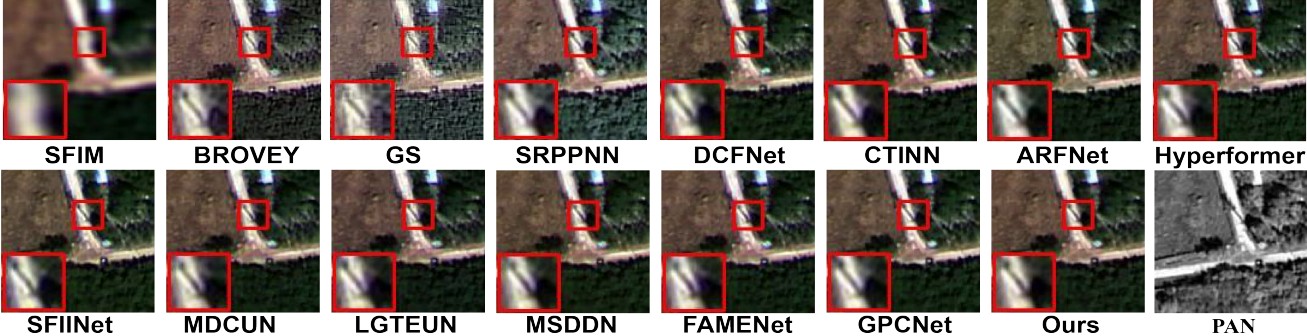

*Figure 7.* Visual comparison on the GaoFen-2 full-resolution scene, with some details magnified for better comparison.

## E. Complexity Analysis

The time complexity of aggregating all basic spatial-spectral relationship patterns is $O(\mathbb{N}n^2)$, and the time complexity of graph convolution is $O(n^2 dl + nmd + nd^2(l-1))$, so the total time complexity of local-wise aggregation is $O(n^2(\mathbb{N} + dl) + nmd + nd^2(l-1))$. The global-wise aggregation first calculates the global-wise spatial-spectral relationship pattern similarity matrix, and the time complexity is $O(n^2\mathbb{N})$. The time complexity of the graph convolution is also $O(n^2 dl + nmd + nd^2(l-1))$, and thus the overall time complexity is $O(n^2(\mathbb{N} + dl) + nmd + nd^2(l-1))$. Lastly, the time complexity of contrastive

| Method | GaoFen-2 | | | WorldView-3 | | | QuickBird | | |
|---|---|---|---|---|---|---|---|---|---|
| | $D_\lambda\downarrow$ | $D_s\downarrow$ | QNR↑ | $D_\lambda\downarrow$ | $D_s\downarrow$ | QNR↑ | $D_\lambda\downarrow$ | $D_s\downarrow$ | QNR↑ |
| SFIM | 0.082 | 0.159 | 0.821 | 0.137 | 0.421 | 0.683 | 0.219 | 0.496 | 0.672 |
| BROVEY | 0.138 | 0.261 | 0.739 | 0.136 | 0.417 | 0.689 | 0.226 | 0.516 | 0.651 |
| GS | 0.074 | 0.246 | 0.703 | 0.141 | 0.407 | 0.676 | 0.231 | 0.526 | 0.632 |
| SRPPNN | 0.077 | 0.116 | 0.817 | 0.128 | 0.319 | 0.757 | 0.159 | 0.362 | 0.700 |
| DCFNet | 0.073 | 0.115 | 0.839 | 0.092 | 0.277 | 0.774 | 0.198 | 0.332 | 0.723 |
| CTINN | 0.072 | 0.114 | 0.834 | 0.072 | **0.114** | **0.834** | 0.133 | 0.338 | 0.732 |
| ARFNet | 0.081 | 0.129 | 0.817 | 0.230 | 0.372 | 0.709 | 0.210 | 0.397 | 0.652 |
| SFIINet | 0.068 | 0.112 | 0.847 | 0.094 | 0.271 | 0.706 | 0.141 | 0.337 | 0.732 |
| Hyperformer | 0.075 | 0.119 | 0.823 | 0.222 | 0.357 | 0.775 | 0.201 | 0.420 | 0.653 |
| MDCUN | 0.065 | 0.110 | 0.850 | 0.088 | 0.257 | 0.783 | 0.138 | 0.339 | 0.730 |
| LGTEUN | 0.063 | 0.121 | 0.841 | 0.091 | 0.254 | 0.785 | 0.137 | 0.336 | 0.737 |
| MSDDN | 0.069 | 0.112 | 0.858 | 0.149 | 0.353 | 0.710 | 0.149 | 0.353 | 0.710 |
| FAMENet | 0.071 | 0.153 | 0.821 | 0.094 | 0.259 | 0.779 | 0.141 | 0.339 | 0.730 |
| GPCNet | 0.072 | 0.113 | 0.820 | 0.116 | 0.312 | 0.759 | 0.144 | 0.338 | 0.726 |
| **Ours** | **0.061** | **0.107** | **0.860** | **0.085** | 0.252 | 0.792 | **0.130** | **0.328** | **0.742** |

*Table 6.* The average results on the real-world full-resolution scenes from GaoFen-2, WorldView-3, and QuickBird datasets. The best results are highlighted in boldface, the second best one is underlined.

learning is $O(n^2d)$. Therefore, the total time complexity of our HetSSNet is $O((\mathbb{N} + dl)n^2 + (m + d^2(l-1))n)$.

| Method | PSNR ↑ | SSIM ↑ | SCC ↑ | SAM ↓ | ERGAS ↓ |
|---|---|---|---|---|---|
| SFIM | 21.415 | 0.542 | 0.721 | 0.115 | 8.855 |
| BROVEY | 22.506 | 0.547 | 0.733 | 0.571 | 8.233 |
| GS | 28.417 | 0.693 | 0.812 | 0.102 | 6.799 |
| SRPPNN | 30.304 | 0.918 | 0.956 | 0.078 | 3.188 |
| DCFNet | 30.624 | 0.924 | 0.958 | 0.072 | 3.092 |
| CTINN | 31.856 | 0.952 | 0.960 | 0.066 | 2.742 |
| SFIINet | 30.597 | 0.924 | 0.956 | 0.074 | 3.080 |
| Hyperformer | 29.943 | 0.913 | 0.945 | 0.081 | 3.426 |
| MDCUN | 31.299 | 0.953 | 0.956 | 0.066 | 2.930 |
| BiMPan | 30.419 | 0.941 | 0.952 | 0.069 | 2.912 |
| LGTEUN | 32.219 | 0.955 | 0.950 | 0.061 | 2.629 |
| MSDDN | 30.850 | 0.926 | 0.955 | 0.073 | 2.995 |
| FAMENet | 30.990 | 0.929 | 0.954 | 0.070 | 2.953 |
| GPCNet | 30.543 | 0.922 | 0.955 | 0.077 | 3.105 |
| Ours | **32.589** | **0.958** | **0.963** | **0.059** | **2.524** |

*Table 7.* Comparison of different methods on eight-band WorldView-3 dataset

The input of local-wise spatial-spectral relationship pattern aggregation includes matrices composed of various basic spatial-spectral relationship patterns and attribute matrix $U \in \mathbb{R}^{n\times m}$, and the weight vector and matrices contain $\alpha_{1:\mathbb{N}}, \mathbb{W}^1_{Local} \in \mathbb{R}^{m\times d}$ and $\mathbb{W}^l_{Local} \in \mathbb{R}^{d\times d}$, and thus the space complexity of local-wise spatial-spectral relationship pattern aggregation is $O(n^2\mathbb{N} + nm + md + d^2(l-1) + \mathbb{N})$. Similarly, the input of the global-wise spatial-spectral relationship pattern aggregation includes matrix $B \in \mathbb{R}^{n\times\mathbb{N}}$ and adjacency matrix $\tilde{\mathbf{A}}_{Global} \in \mathbb{R}^{n\times n}$, and the weight vector and matrices contain $\beta_{1:\mathbb{N}}, \mathbb{W}^1_{Global} \in \mathbb{R}^{m\times d}$ and $\mathbb{W}^l_{Global} \in \mathbb{R}^{d\times d}$, and thus the space complexity of global-wise spatial-spectral relationship pattern aggregation is $O(n\mathbb{N} + n^2 + md + d^2(l-1) + \mathbb{N})$. Therefore, the total space complexity of HetSSNet is $O(\mathbb{N}n^2 + (m + \mathbb{N})n + md + d^2(l-1) + \mathbb{N})$.

