# OpenReview forum: "HetSSNet: Spatial-Spectral Heterogeneous Graph Learning Network for Panchromatic and Multispectral Images Fusion"
_ICML.cc/2025/Conference — ICML 2025 poster_

### Official Review · Reviewer_bvrf · 2025-02-24

**Overall Recommendation:** 4

**Summary:**

This paper proposes a spatial-spectral heterogeneous graph learning network for panchromatic and multispectral images fusion. Specifically, the authors explore the pansharpening-specific relationships in the heterogeneous graph structure. Then, they extract the multiple relationship patterns by the designed basic relationship pattern aggregation module. Finally, the unified spatial-spectral representation across different relationships is learned from local and global perspectives.

## update after rebuttal
My concerns have been addressed and would like to recommend accept.

**Claims And Evidence:**

Yes

**Essential References Not Discussed:**

None

**Experimental Designs Or Analyses:**

Yes. The experimental designs including baseline comparisons and ablation studies are coherent. No major flaws or inconsistencies are found in the experimental setup or analyses.

**Methods And Evaluation Criteria:**

Yes

**Other Comments Or Suggestions:**

There are some inaccurate expressions in the manuscript—particularly around lines 35 and 200. The authors should carefully review and correct these sections.

**Other Strengths And Weaknesses:**

Strengths
1. The authors design the first spatial-spectral heterogeneous graph structure for remote sensing pansharpening.
2. Building on the spatial-spectral heterogeneous graph structure, the authors further develop both local and global relationship pattern aggregation mechanisms.
3. The paper is well-organized and clearly written.
Weaknesses:
1. In the discussion of the graph construction challenge at line 60, the authors highlight the spectral relationship between LR-MS and PAN images. Why do they not also consider the spatial relationship between these images?
2. What are the advantages of the heterogeneous graph learning mechanism designed by the author compared with the methods used in previous heterogeneous graph learning networks (e.g., MHGCN)?
3. The roles of local- and global-level aggregations warrant further analysis, and the authors should clarify how these two aspects are balanced.
4. Pansharpening datasets also have some eight-band remote sensing images. Can the authors provide experimental comparison of eight-band datasets?
5. Graph structure typically demand substantial computational resources when dealing with large-scale graphs or high-dimensional node features. Therefore, the authors should compare the computational cost of their proposed approach with that of existing methods.

**Questions For Authors:**

Please refer to the Weakness.

**Relation To Broader Scientific Literature:**

The previous method [1] only uses GCN to model global relationships in Euclidean space. The authors pioneer spatial-spectral heterogeneous graph structure in non-European space for remote sensing pansharpening.
[1] When pansharpening meets graph convolution network and knowledge distillation. IEEE Transactions on Geoscience and Remote Sensing.

**Theoretical Claims:**

Yes. The arguments and derivations about the heterogeneous graph architecture related to spatial-spectral relationships appear logically sound.

---

> ### Author Rebuttal · Authors · 2025-04-01
>
> Thank you for your thoughtful and encouraging review. The point-to-point answer is provided below.
>
> **A1.** We analyze the spatial relationship of a large number of PAN/LR-MS/target HR-MS (GT) image pairs in the Appendix. As can be seen from Fig. 3 in the Appendix, the spatial histogram of each spectral band of LR-MS is extremely different from that of each spectral band of GT. In contrast, the spatial histogram of PAN is very close to that of each spectral band of GT. Therefore, we can obtain spatial property-related relationship prior, i.e., modeling only the spatial relationship of PAN images can reconstruct the required spatial property of target HR-MS images.
>
> **A2.** The designed spatial-spectral heterogeneous graph is regarded as the attributed multiplex heterogeneous graph. Existing methods do not explicitly model multiplex heterogeneous structures, but rather regard them as a linear superposition of relations. For example, MHGCN realizes message passing in multiplex heterogeneous networks by automatically capturing useful relation-aware meta-paths. It is only regarded as a simple superposition of relations, while ignoring the learning of semantic information. But in fact, when learning the spatial-spectral heterogeneous graph, we should not simply regard it as a linear superposition of simple individuals, but should focus on its own high-level semantic information. The proposed method can express rich high-level semantic information by extracting basic spatial-spectral relationship patterns and multi-level relationship aggregation. The ablation experiments in the manuscript (Tab.3) also demonstrate the effectiveness of our learning mechanism.
>
> **A3.**  In this paper, we use the contrastive learning loss (InfoNCE loss [1]) to balance the local-wise and global-wise aggregation branch, as denoted in Eq.8. In Eq.8, the positive sample pairs are the numerator, i.e., $\exp\mathrm{(}\boldsymbol{s}(\mathbf{H} _{\boldsymbol{Local},i},\mathbf{H} _{\boldsymbol{Global},i}))$, which enforces the local features to be similar to their corresponding global features; The negative sample pairs are denominator, i.e., $\sum _{j\in \mathrm{V}}{\exp}(\boldsymbol{s}(\mathbf{H} _{\boldsymbol{Local},i},\mathbf{H} _{\boldsymbol{Global},j}))$, which contains the global features of all samples ($j\ne i$). Negative sample pairs force the model to learn discriminative local features while requiring the global features to be sufficiently discriminative. The temperature coefficient $\tau$ in Eq.8 controls the weight of local-wise and global-wise branches.  In the early stages of training, the model may focus more on local details. As training progresses, global features gradually dominate, guiding local features to learn semantic consistency.
>
> **A4.**  The experimental results of eight-band WorldView-3 dataset are shown in the table as below. We will add these results in final version.
>
> | Methods    | PSNR ↑ | SSIM ↑ | SCC ↑ | SAM ↓ | ERGAS ↓ |
> |------------|--------|--------|-------|-------|---------|
> | SFIM       | 21.415 | 0.542  | 0.721 | 0.115 | 8.855   |
> | BROVEY     | 22.506 | 0.547  | 0.733 | 0.571 | 8.233   |
> | GS | 28.417 | 0.693  | 0.812 | 0.102 | 6.799  |
> | SRPPNN| 30.304 | 0.918  | 0.956 | 0.078 | 3.188   |
> | DCFNet     | 30.624 | 0.924  | 0.958 | 0.072 | 3.092   |
> | CTINN      | 31.856 | 0.952  | 0.960 | 0.066 | 2.742   |
> | SFIINet    | 30.597 | 0.924  | 0.956 | 0.074 | 3.080   |
> | Hyperformer | 29.943 | 0.913 | 0.945 | 0.081 | 3.426 |
> | MDCUN      | 31.299 | 0.953  | 0.956 | 0.066 | 2.930   |
> | BiMPan      | 30.419 | 0.941  | 0.952 | 0.069 | 2.912  |
> | LGTEUN     | 32.219 | 0.955  | 0.950 | 0.061 | 2.629   |
> | MSDDN      | 30.850 | 0.926  | 0.955 | 0.073 | 2.995   |
> | FAMENet    | 30.990 | 0.929  | 0.954 | 0.070 | 2.953   |
> | GPCNet      | 30.543 | 0.922  | 0.955 | 0.077 | 3.105   |
> | **Ours**   | **32.589** | **0.958** | **0.963** | **0.059** | **2.524** |
>
> **A5.** We have calculated the space complexity of our proposed model in the manuscript (Line 844-851). As shown in the table below, we present computational cost of our method and all comparison methods.
>
> | Methods   | SRPPNN | DCFNet | CTINN | SFIINet | Hyperformer | MDCUN | BiMPan | LGTEUN | MSDDN |FAMENet | GPCNet | Ours |
> |---|--------|--------|-------|-------|---------|--------|--------|--------|---------|-------|---------|---------|
> |Params|1.711| 2.905 | 0.039 | 0.085 | 43.938 |0.098| 0.396  | 0.202| 0.011 | 0.091| 0.087| 0.167|
>
> **A6.**  Thank you for pointing it out. We will carefully correct the descriptions in the Line 35 and 200.
>
> [1] Chen T,  et al. A simple framework for contrastive learning of visual representations. ICML, 2020.

---

### Official Review · Reviewer_LVzz · 2025-03-04

**Overall Recommendation:** 5

**Summary:**

For pansharpening, recent modeling frameworks are relatively rigid and have limitations when dealing with irregular ground objects in remote sensing images. To address this issue, this paper proposes a spatial-spectral heterogeneous graph learning network named HetSSNet. It constructs a heterogeneous graph structure that explicitly describes the specific relationships between PAN and MS images. A basic relationship pattern generation module is designed to extract multiple fundamental spatial-spectral relationship patterns from the constructed heterogeneous graph. To automatically capture local information and global relevant information across different complex relationship pattern between nodes, it designs the relationship pattern aggregation module from the local and global perspective. The experimental results demonstrate that the model achieves satisfactory performance on multiple datasets.

**Claims And Evidence:**

The methods and experimental design of the paper are clearly presented and well-structured.

**Essential References Not Discussed:**

I have not seen any similar spatial-spectral heterogeneous graph structure for remote sensing pansharpening. The paper is therefore innovative in this regard.

**Experimental Designs Or Analyses:**

The authors conduct ablation experiments to validate the rationality of the Basic Relationship Pattern Generation module and the Relationship Pattern Aggregation module.

**Methods And Evaluation Criteria:**

The datasets used are classic in the field of pansharpening, and the full-resolution and reduced-resolution experiments are also commonly conducted in this field.

**Other Comments Or Suggestions:**

Please refer to the Weaknesses.

**Other Strengths And Weaknesses:**

Strengths:
1. The article is well-structured and logically coherent, progressing systematically from problem statement to method design and experimental validation.
2. The authors analyze pansharpening data in the appendix section, and obtain the priors of task-related spatial-spectral relationships. The paper proposes a novel spatial-spectral heterogeneous graph to model these relationships in a more flexible way.
3. The experimental design is rigorous, utilizing three widely recognized datasets and incorporating both comparative and ablation studies for validation.
4. The proposed method extracts multiple basic relationship matrices based on edge types and aggregates these basic relationships from local and global perspectives to obtain a unified spatial-spectral relationship representation.

Weaknesses:
1. The authors says that recent CNN/Transformer is rigid in modeling relationships in pan-sharpening. What does this "rigidity" mean in remote sensing image? Further discussion is recommended.
2. In definition 2, the authors give the definition of the basic relationship pattern. It is recommended that the authors give the example to illustrate the basic relationship pattern in the constructed spatial-spectral heterogeneous graph.
3. The related work needs improvement. Personally, in the current version, it should express what are the current mechanisms for learning heterogeneous graph.

**Questions For Authors:**

Please refer to the Weaknesses.

**Relation To Broader Scientific Literature:**

The paper provides examples and introduces CNN-based and Transformer-based methods, such as DCFNet and CTINN, pointing out that their modeling is overly rigid. To address this issue, the paper introduces its innovative approach.

**Theoretical Claims:**

The authors could further explain why the method based on CNN/Transformer is rigid and why the method based on heterogeneous graphs is more flexible.

---

> ### Author Rebuttal · Authors · 2025-03-31
>
> Thank you for your thoughtful and encouraging review. The point-to-point answer is provided below.
>
> **A1.** The convolution kernel of CNN slides on a regular grid and cannot adapt to the curved, broken, radial and other remote sensing image features. The core of Transformer is global self-attention, and fixed-size patch splitting (such as 16$\times$16) will break the structure of irregular features (such as rivers and roads). In fact, the rigidity of CNN/Transformer comes from its regular modeling assumption of Euclidean space, which is inconsistent with the non-Euclidean data characteristics of remote sensing image features (such as mountains, rivers, etc.).
>
> **A2.** We give two examples of the base relationship pattern. For example, $\mathbf{P}\xrightarrow{spectral}\mathbf{L}$ is a basic relationship pattern between PAN and LR-MS nodes, which represents that there is only spectral interaction between the PAN node and the LR-MS node . $\mathbf{P}\xrightarrow{spatial/spectral}\mathbf{L}$ is also a basic relationship pattern, where there are spatial interaction and cross-modal spectral interaction between the PAN node and the LR-MS node.
>
> **A3.**   We will add related work of heterogeneous graph learning strategies in Sec. 2.2. For example, GATNE [1] learns the representation of nodes in each specific relationship, respectively, and then aggregate them to obtain the final node representation. DualHGNN [2] introduces two hypergraphs into the representation learning of multiplex heterogeneous networks, and learns node embedding using spectral hypergraph convolutional networks. FAME [3] realizes message passing in multiplex heterogeneous networks by automatically capturing useful relation-aware meta-paths.
>
> [1] CEN Y, et al. Representation Learning for Attributed Multiplex Heterogeneous Network. SIGKDD, 2019.
>
> [2] XUE H, et al. Multiplex Bipartite Network Embedding using Dual Hypergraph Convolutional Networks. arXiv, 2021.
>
> [3] LIU Z, et al. Fast Attributed Multiplex Heterogeneous Network Embedding. CIKM, 2020.

---

> > ### Comment · Reviewer_LVzz · 2025-04-02
> >
> > Thank you for the response. All my comments have been well addressed, I have no further questions. I believe this paper offers valuable insights to the community, with clear contributions, and I therefore decide to upgrade my rating to 5.

---

### Official Review · Reviewer_XsAf · 2025-03-11

**Overall Recommendation:** 3

**Summary:**

In this paper, authors propose a spatial-spectral heterogeneous graph learning network, termed as HetSSNet. Specifically, it segments each band of the pansharpening data into non-overlapping patches. By leveraging graph structures, it captures intrinsic relationships among these patches. Furthermore, it employs contrastive learning to enrich feature representations and enhance the overall performance of pan-sharpening.

**Claims And Evidence:**

Yes.

**Essential References Not Discussed:**

No, all related pan-sharpening methods based on graph neural networks have been discussed.

**Experimental Designs Or Analyses:**

Yes, the experiment complies with the general settings.

**Methods And Evaluation Criteria:**

Yes.

**Other Comments Or Suggestions:**

[-] In Section 3.1, the symbol "U" in the defined graph $G(V,E,U)$ is not denoted.

[-] In Section 2.2, the description "Non-European" appears inappropriate or unclear.

[-] In line 176 and line 182, the width and height of LRMS image should not be assumed to be equal to those of PAN image.

[-] The distinguishability of the symbols should be improved for clarity.

[-] In line 205, $\tilde{A}_l$ lacks a clear definition.

**Other Strengths And Weaknesses:**

### Strengths:
[+] Based on statistical analysis, the authors constructed a graph neural network incorporating a spatial-spectral heterogeneous graph construction module, a basic relationship pattern generation module, and a relationship pattern aggregation module to enhance the performance of pansharpening.

[+] The authors theoretically analyzed the computational complexity of the proposed method.

[+] The authors utilize graph structures to capture long-range dependencies within images, which is a relatively novel approach in the field of pansharpening.

### Weaknesses:
[-] The authors claim that the method is non-Euclidean but abandon this concept in the implementation, making the theoretical and intuitive justification unconvincing.

[-] In Section 3.3, the authors mention, "We divide the image into N overlapping patches and transform each patch of the PAN image into a feature vector." However, the details of this transformation are unclear.

[-] The ablation experiments in the paper are insufficient. Specifically, in experimental settings (c) and (d), the contrastive loss cannot be applied due to the absence of positive and negative sample pairs. After combining the two modules, the authors do not adequately discuss the role of the contrastive loss, both qualitatively and quantitatively.

[-] The running efficiency, number of parameters, and floating-point operations per second (FLOPs) of the method are not reported.

[-] The relationship between the final node representation $H$ and the network output is not clearly explained.

[-] The article employs comparative methods that are somewhat outdated, focusing primarily on approaches developed before 2023. Additionally, the references need to be updated to include more recent research and developments in the field.

**Questions For Authors:**

Please refer to the weakness.

**Relation To Broader Scientific Literature:**

The main contribution of the paper is the introduction of the graph neural network structure in the field of pan-sharpening. This provides insights into improving the network structure in related fields. However, there are not many related articles in recent years, and it may be widely explored by the community in the future.

**Theoretical Claims:**

No, there is no theoretical proof in the article.

---

> ### Author Rebuttal · Authors · 2025-03-31
>
> Thank you for your thoughtful and encouraging review. The point-to-point answer is provided below.
>
> **A1.** The non-Euclidean characteristics of the method are reflected in two aspects: the data structure and relationship modeling. **(a) Spatial-spectral heterogeneous graph construction.** We construct the spatial-spectral heterogeneous graph with two types of nodes (PAN and LR-MS nodes) and heterogeneous edges modeling complex spatial-spectral relationships . The graph topology is dynamically constructed via an adaptive $k$-nearest neighbor algorithm (Line 190-206). Thus, the defined spatial-spectral heterogeneous graph inherently represents a non-Euclidean data structure. **(b) Non-Euclidean relationship modeling.** We design relationship pattern aggregation module based on the non-Euclidean learning operation (i.e., GCN) (Sec. 3.5). It explicitly respect the graph topology, avoiding the limitations of Euclidean assumptions.
>
> **A2.** The convolution operation is applied to the PAN image patches and obtain the corresponding feature vector.
>
> **A3.** The experimental setting (c) and (d) could verify the role of local features and global features in contrastive learning. In fact, Setting (c) and (d) individually makes Eq.8 degenerate into self-contrast learning with only global/local features. The results show that our cross-view local-global contrastive learning is more effective. We add two sets of ablation experiments: visual ablation experiment (https://anonymous.4open.science/r/kk3/exp.png), and the ablation experiment of the temperature coefficient $\tau$ in Eq.8 as follows.
>
> |$\tau$|PSNR↑|SAM↓|SSIM↑|QNR↑|$D_{s}$↓|
> |--|--|--|--|--|--|
> |0.01|48.577|0.025|0.985|0.849|0.118|
> | **0.1** |**49.541** |**0.015**|**0.994**|**0.860**|**0.107**|
> |0.5|49.311|0.019|0.990|0.856|0.109|
> |1.0|48.011|0.029|0.976|0.840|0.125|
>
> **A4.** We present the inference time, number of parameters, and FLOPs of our method. We will add them in the final version.
>
> | Metric |Time (s/img) | Params (M) | FLOPs (G) |
> |--------------|-------|-------|------|
> | HetSSNet| 0.0327|0.167| 1.927|
>
> **A5.** The final node representation $\mathbf{H}$ is first mapped to the target dimension through the fully-connected layer, and then refined using the convolutional operation to generate the final HR-MS image. We will add the corresponding description after Eq.9.
>
> **A6.** We add three comparison methods published in 2024, i.e., FusionMamba [1], WINet [2] and HFIN [3], as shown in tables below, these comparison methods will be updated in the final version. In addition, we will update the references in the Sec.1 and Sec.2.1.
>
> **WV-3:**
> | Method|PSNR↑|SSIM↑|SAM↓|ERGAS↓|SCC↑|$D_{\lambda}$|$D_{s}$↓|QNR↑|
> |-|-|-|-|-|-|-|-|-|
> |FusionMamba|38.727|0.872|0.069|5.437|0.934|0.097|0.261|0.752|
> |WINet|39.781|0.887|0.056 |4.769|0.969|0.132|0.331| 0.740|
> |HFIN|32.126|0.837|0.083|6.627|0.837|0.142|0.341|0.723|
> |**Ours**|40.623|0.905|0.054|4.741|0.972|0.085|0.252|0.792|
>
> **QB:**
> |Method|PSNR↑|SSIM↑|SAM↓|ERGAS↓|SCC↑|$D_{\lambda}$↓|$D_{s}$↓|QNR↑|
> |-|-|-|-|-|-|-|-|-|
> | FusionMamba|34.021|0.881|0.047|2.931|0.904|0.152|0.381|0.697|
> |WINet|37.209|0.927|0.032|2.212|0.931|0.142|0.341|0.714|
> |HFIN|33.616|0.811|0.050|3.174|0.869|0.211|0.376|0.662 |
> |**Ours**|37.228|0.931|0.029|2.022|0.951|0.130|0.328|0.742|
>
> **GF-2:**
> |Method|PSNR↑|SSIM↑|SAM↓|ERGAS↓|SCC↑|$D_{\lambda}$↓|$D_{s}$↓|QNR↑|
> |-|-|-|-|-|-|-|-|-|
> |FusionMamba|46.242|0.976| 0.027|1.278|0.972|0.074|0.121|0.756|
> |WINet|49.405| 0.984 |0.016| 0.968|0.977|0.060|0.108|0.855|
> |HFIN|41.938|0.966 |0.039|2.534|0.894|0.136|0.158|0.722|
> |**Ours**|49.541|0.994|0.015| 0.955|0.990|0.061|0.107|0.860|
>
> **A7.** The matrix $\mathrm{U}$ is constructed by stacking the feature vectors of all nodes in the spatial-spectral heterogeneous
> graph, where each row is the associated node feature vector of node $v_i$ (Line 150-152). For example, the corresponding feature vectors of PAN images/ LR-MS images patches are used to form the matrix $\mathrm{U}$ (Line 214).
>
> **A8.**  "Non-European" is a typo, it should be "Non-Euclidean". We will correct it in Line 135.
>
> **A9.** During the implementation for the pansharpening models, the LR-MS needs to be upsampled to the size of PAN before fusion [2]. In addition, the dimension definition of LR-MS follows the published pansharpening work [2], thus, in method section, it is reasonable to define it as the same length and width as the PAN image.
>
> **A10.**  Thanks for your suggestion, we will correct them.
>
> **A11.**  It is a typo, it should be "$\tilde{\mathbf{A}}_{\boldsymbol{Local}} $ " generated from Eq.1. We will correct it in Line 205.
>
> [1] Peng S, et al. Fusionmamba: Efficient remote sensing image fusion with state space model. TGRS, 2024.
>
> [2] Tan J, et al. Revisiting Spatial-Frequency Information Integration from a Hierarchical Perspective for Panchromatic and Multi-Spectral Image Fusion. CVPR, 2024.
>
> [3] Zhang J, et al. Pan-Sharpening With Wavelet-Enhanced High-Frequency Information. TGRS, 2024.

---

> > ### Comment · Reviewer_XsAf · 2025-04-05
> >
> > The authors provide detailed responses to each of the raised issues, and incorporate a substantial number of experiments. These can address my concerns and simultaneously improve the quality of the paper. Furthermore, taking into consideration the opinions of other reviewers, I will raise my rating.

---

### Official Review · Reviewer_UFR7 · 2025-03-13

**Overall Recommendation:** 4

**Summary:**

Remote sensing pansharpening involves fusing panchromatic (PAN) images with low-resolution multi-spectral (LR-MS) images to produce high-resolution multi-spectral (HRMS) images. Traditional methods like CNN and Transformer treat images as grids of pixels in Euclidean space, which struggle with irregular ground objects in remote sensing. Graphs offer a more flexible structure but face challenges in modeling spatial-spectral properties: 1) creating a customized graph structure for spatial-spectral relationships, and 2) learning a unified spatial-spectral representation. To overcome these, the authors propose HetSSNet, a spatial-spectral heterogeneous graph learning network. HetSSNet constructs a heterogeneous graph tailored for pansharpening, extracts multiple relationship patterns, and aggregates these patterns to learn a unified spatial-spectral representation from both local and global perspectives. Extensive experiments show that HetSSNet outperforms existing methods and generalizes well.

**Claims And Evidence:**

Yes

**Essential References Not Discussed:**

It is recommended that the author should discuss the GCN-based panchromatic sharpening method, GPCNet, published in TGRS2024.

**Experimental Designs Or Analyses:**

The experimental design is reasonable and provides competitive results on reduced-and full-resolution datasets.

**Methods And Evaluation Criteria:**

In order to solve the rigid modeling of CNN/Transformer, the author introduces the heterogeneous graph to establish complex spatial-spectral relationships, this approach is interesting and novel. Experiments on three pansharpening datasets demonstrate consistent improvements over CNN-based/Transformer-based methods, proving that the proposed method has the potential to break through the limitation of the current modeling framework.

**Other Comments Or Suggestions:**

1. The table should be expanded to include the sources or origins (provenance) of the different methods used in the comparison.

2. Avoid using abbreviations like "i.e." in the abstract and conclusion sections.

3. In section 3.6, it contains a typo ("Optimization " instead of " optimization ").

**Other Strengths And Weaknesses:**

Strengths ：

1. The paper analyzes the two spatial-spectral relationship priors in the pansharpening, providing a good basis for subsequent modeling.

2. The paper uses heterogeneous graph to construct complex and diverse spatial-spectral relationships between LR-MS and PAN images, this is an interesting solution. In addition, to learn unified spatial-spectral representation, the proposed method extracted base relationship pattern and learn them from the local and global perspective.

3. Comprehensive experiments on WorldView-3, GaoFen-2 and QuickBird datasets demonstrate consistent improvements over state-of-the-art pansharpening methods, validated by ablation studies analyzing individual HetSSNet components.

4. Unlike the other CNN/Transformer-based methods, the proposed method attempts to design a new pansharpening modeling framework in non-Euclidean space. This study shows a commendable beginning.

5. The article is well written and the figures are clear.

Weakness:

1. Why do CNN and Transformer structures lead to redundancy and also do Mamba or Diffusion structures suffer from the above drawbacks? Please provide a more in-depth explanation of the following point:‘’Since these ground objects are usually not quadrate whose shape is irregular, the commonly used grid structures in the modeling architecture like CNN and Transformer are redundant and inflexible to process them “.

2. In the Local-wise Aggregation, the authors used only a single-layer GCN to learn node representations with interaction information. The motivation for this approach is not clearly explained.

3. The author should discuss the GCN-based pansharpening method (i.e., GPCNet[1]) in the introduction section to make it clearer how it differs from this proposed method. In addition, it is recommended to add the analysis of recent heterogeneous graph learning strategies (e.g., meta-path sampling) in the related work section.

4. This paper uses the basic relationship modeling strategy. Meta-path sampling is a commonly used heterogeneous graph modeling mechanism. Why not use the mete-path sampling mechanism? What are the advantages of the learning strategy used in this paper?

5. The author should add the experimental results of the eight-band pansharpening dataset.

**Questions For Authors:**

In the section on Pansharpening-specific relationship priors, the comparison of spectral band distributions is currently limited to neighboring spectra, such as band1 and band2. Could the analysis be extended to explore distribution comparisons across broader spectral ranges, for example, between band1 and band3, or band1 and band4?

**Relation To Broader Scientific Literature:**

Recent learning-based methods often rely on the Euclidean space assumption, which may not fully capture the complex geometric structure of remote sensing data. The non-Euclidean framework proposed in this paper draws on the progress of graph, which have shown potential in dealing with irregular data. This contribution extends the application of non-Euclidean methods to the field of pansharpening, addressing the urgent need for more flexible and adaptive spatial-spectral modeling.

**Theoretical Claims:**

The spatial-spectral relationship priors for the pan-sharpening task are verified and analyzed on a large amount of pansharpening data.

---

> ### Author Rebuttal · Authors · 2025-03-31
>
> Thank you for your thoughtful and encouraging review. The point-to-point answer is provided below.
>
> **A1.** The convolution kernel of CNN assumes that local features are spatially invariant, but the morphology of objects (such as rivers and roads) may change nonlinearly with the terrain. For example, the curvature of mountain roads is different from that of plains. It is difficult for a fixed-size convolution kernel to adaptively adjust the receptive field and may perform redundant calculations on irrelevant areas (such as the background). Transformer captures long-range dependencies through self-attention, but the correlation of objects in remote sensing images often has local-global heterogeneity (such as regular local structures of buildings, but random distribution of vegetation). Global calculations introduce noise from irrelevant pixels and are computationally expensive. Mamba and Diffusion also have the problem of rigid modeling. Specifically, Mamba's default scanning path (such as from left to right, from top to bottom) is not suitable for anisotropic objects (such as diagonal roads or radial urban layouts). Diffusion is due to the denoising network (usually U-Net) still relying on the local inductive bias of convolution.
>
> **A2.** We use two-layer GCN. In fact, the over-smoothing problem limits the GCN model from stacking more aggregation layers. The purpose of this work is not to solve the over-smoothing problem of GCN, but to use GCN with a suitable number of layers to learn the designed spatial-spectral heterogeneous graph. Ablation experiments in the manuscript have shown that even a small number of GCN layers can achieve promising performance (Tab. 4).
>
> **A3.** GPCNet uses the full-graph modeling strategy, in which all nodes and edges are of the same type and all nodes have relationships. Pansharpening involves the fusion of PAN and LR-MS, which have heterogeneous characteristics. Therefore, the full-graph structure is sub-optimal for pansharpening. The designed spatial-spectral heterogeneous graph can explicitly model the heterogeneity of PAN and LR-MS, flexibly fuse spatial-spectral relationships, and adapt to local and global modeling requirements.  In addition, we will add the analysis of recent heterogeneous graph learning strategies in the final version (please refer to **A4**).
>
> **A4.**  Meta-path sampling requires pre-setting of fixed paths and cannot flexibly adapt to the spectral-spatial coupling of different regions; it requires sampling all possible paths, which has high computational complexity (especially for long paths); the path's semantics are single, making it difficult to integrate the multi-objective constraints of spectral fidelity and spatial enhancement. However, we use the base relationship pattern strategy to model multiple edge types at the same time and dynamically weight the relationships between different relationships. The ablation model (Modal (a) in Tab.3) verifies this in the manuscript.
>
> **A5.**  The experimental results of the eight-band worldview-3 dataset are shown in the table below, which shows that our method could achieve promising results on the eight-band dataset. We will add these experimental results in the final revision.
>
> | Methods    | PSNR ↑ | SSIM ↑ | SCC ↑ | SAM ↓ | ERGAS ↓ |
> |------------|--------|--------|-------|-------|---------|
> | SFIM       | 21.415 | 0.542  | 0.721 | 0.115 | 8.855   |
> | BROVEY     | 22.506 | 0.547  | 0.733 | 0.571 | 8.233   |
> | GS | 28.417 | 0.693  | 0.812 | 0.102 | 6.799  |
> | SRPPNN| 30.304 | 0.918  | 0.956 | 0.078 | 3.188   |
> | DCFNet     | 30.624 | 0.924  | 0.958 | 0.072 | 3.092   |
> | CTINN      | 31.856 | 0.952  | 0.960 | 0.066 | 2.742   |
> | SFIINet    | 30.597 | 0.924  | 0.956 | 0.074 | 3.080   |
> | Hyperformer | 29.943 | 0.913 | 0.945 | 0.081 | 3.426 |
> | MDCUN      | 31.299 | 0.953  | 0.956 | 0.066 | 2.930   |
> | BiMPan      | 30.419 | 0.941  | 0.952 | 0.069 | 2.912  |
> | LGTEUN     | 32.219 | 0.955  | 0.950 | 0.061 | 2.629   |
> | MSDDN      | 30.850 | 0.926  | 0.955 | 0.073 | 2.995   |
> | FAMENet    | 30.990 | 0.929  | 0.954 | 0.070 | 2.953   |
> | GPCNet      | 30.543 | 0.922  | 0.955 | 0.077 | 3.105   |
> | **Ours**   | **32.589** | **0.958** | **0.963** | **0.059** | **2.524** |
>
> **A6.**  Thanks for your suggestion, we will add sources of the comparison methods in the Tab.2 and Tab.6.
>
> **A7.**  Thanks for your suggestion, we will remove these abbreviations and recalibrate correspond description in the final version.
>
> **A8.** Thank you for pointing it out, we will correct it.

---

> > ### Comment · Reviewer_UFR7 · 2025-04-07
> >
> > Thank you for your detailed response, the author has addressed my concerns and I recommend acceptance.

---

### Decision · Program_Chairs · 2025-05-01

**Decision:**

Accept (poster)

**Comment:**

After the author-reviewer discussion stage, all reviewers are positive about this submission. AC agrees with the reviewers and recommends acceptance. Given that the authors provided substantial additional supporting materials in the rebuttal, which helped resolve the reviewers' concerns, it is important that these materials are properly incorporated into the camera ready version of the paper.